# Quantitative Assessment of Horizontal Ecological Compensation for Cultivated Land Based on an Improved Ecological Footprint Model: A Case Study of Jiangxi Province, China

**DOI:** 10.3390/ijerph20054618

**Published:** 2023-03-06

**Authors:** Xiaoyong Zhong, Dongyan Guo, Hongyi Li

**Affiliations:** 1Chinese Academy of Natural Resources Economics, Beijing 101149, China; 2School of Tourism and Urban Management, Jiangxi University of Finance and Economics, Nanchang 330000, China

**Keywords:** ecological footprint model, ecosystem service function assessment, cultivated land horizontal ecological compensation, Jiangxi

## Abstract

Cultivated land horizontal ecological compensation is an essential means of reconciling agricultural ecosystem protection and regional economic development. It is important to design a horizontal ecological compensation standard for cultivated land. Unfortunately, there are some defects in the existing quantitative assessments of horizontal cultivated land ecological compensation. In order to raise the accuracy of ecological compensation amounts, this study established an improved ecological footprint model based on the ecosystem service function, focused on estimating the value of ecosystem service function, ecological footprint, ecological carrying capacity, ecological balance index and ecological compensation values of cultivated land in all cities of Jiangxi province. It then analyzed the rationality of ecological compensation amounts in Jiangxi province, which is one of the 13 provinces of major grain-producing areas in China. The results show the following: (1) The total value of soil conservation service function, carbon sequestration and oxygen release service function and ecosystem service function in Jiangxi province showed a spatial distribution trend of “gradually increasing around Poyang Lake Basin”. (2) The cultivated land ecological deficit areas in Jiangxi province are Nanchang City, Jiujiang City and Pingxiang City; ecological surplus areas are Yichun City, Ji’an City and eight other cities; and there is an obvious “Spatial Agglomeration” phenomenon in ecological deficit and ecological surplus areas where ecological deficit areas are mainly concentrated in the northwest region of Jiangxi. (3) The amount needed to attain fair ecological compensation for cultivated land is 5.2 times the payment amount for cultivated land; this indicated there is larger arable land, a favorable condition for agricultural cultivation, and better supply capacity of ecosystem services in most of the cities of Jiangxi. (4) The compensation amount for cultivated land ecological surplus areas in Jiangxi province is generally higher than the cost of ecological protection, and its proportion in GDP, fiscal revenue and agriculture-related expenditure is significantly higher than that in ecological deficit areas; this indicated that the compensation value of cultivated land could play the driving role in the protective behavior for cultivated land. The results provide a theoretical and methodological reference for the construction of horizontal ecological compensation standards for cultivated land.

## 1. Introduction

Payment for ecosystem services (PES) is defined as the ecosystem services of voluntary transactions between suppliers and buyers (Wunder et al., 2008) [1]. In China, ecological compensation has a similar connotation to PES (Pan et al., 2017) [2]. The ecological compensation is classified into vertical payments and horizontal payments based on the source of funds; the subject of former is the Central People’s Government of China, the latter is the local government (Huang, 2015) [3]. The Central People’s Government of China has launched pilot projects of ecological compensation for cultivated land, such as the Grain for Green Program and the fallow system via vertical fiscal transfer payment since the 1990s. However, there exist many problems in executing these projects of ecological compensation, such as overlapping PES schemes, inefficient use of funds and lack of effective monitoring tools [4,5]. Compared with the vertical fiscal transfer payment from top to bottom according to the organization of administrative structures, a horizontal ecological compensation could not only strengthen the role of the market to develop ecological compensation, but also realize the institutional arrangement of internalizing the externalities of ecological protection [6,7,8]. Furthermore, China is devoted to exploring the horizontal ecological compensation based on the market mechanism, which and vertical ecological compensation make up the ecological compensation system to promote ecosystem balance [4,9]. In recent years, horizontal ecological compensation has been applied to basin, water rights trading and pollutant emissions trading, and carbon emissions trading since 2012 [10]. It should be noted that the horizontal ecological compensation framework has not yet been implemented in cultivated land. Despite the criteria of cultivated land ecological compensation being well documented [11,12,13,14], there is no uniform calculation method for the criteria, and the mechanism of horizontal farmland ecological compensation has not been fully set in China [15].

The literature related to cultivated land ecological compensation payment methods has mainly used the following methods: the conditional valuation method (CVM) [15,16,17,18,19,20,21] and the ecological footprint model [22,23]. The CVM is a method of measuring the value of ecosystem service by directly examining the economic behavior of respondents in a hypothetical market through a questionnaire survey to obtain consumers’ willingness to pay for it (Jiang et al., 2022) [24]. The ecological footprint model is a methodology to assess “The total area of productive land required continuously to produce all the resources consumed and to assimilate all the wastes produced” [23]. The latter method is more helpful to reflect ecological benefits for farmland than the CVM when evaluating the value of ecological compensation in current research. However, from the calculation method of the traditional ecological footprint model emerge some controversies, which need to be still improved. Specifically, the theories and methods of accounting ecosystem service value in the ecological footprint model are difficult to unify [25,26,27]. To determine the ecological compensation standards, many studies estimated ecosystem service value for cultivated land using the equivalent factor method and the functional value method of ecosystem services [28,29]. The equivalent factor method is economical, convenient and has a wide applicability. It was originally proposed by Xie Gaodi, a Chinese scholar, based on the classification of ecosystem services by Costanza (Xie et al., 2008) [30], and is widely used in the assessment of ecosystem service value for cultivated land [31,32]; its advantage is providing ecological value assessments at a national scale. However, the fixed coefficient of the equivalent factor could hardly reflect the spatial heterogeneity of different ecosystem service functions because ecosystem service functions are dynamic (Zhang et al., 2021) [27]. Compared with the former, the dynamic change of spatio-temporal heterogeneity in ecosystem services at a regional scale could be reflected more accurately by the functional value method of ecosystem services, which established a production equation between a single service and local ecological environment variables [33,34].

Hence, this study attempted to assess the value of horizontal ecological compensation for cultivated land based on an improved ecological footprint model. First, the cultivated land in Jiangxi province is divided into 880,521 grids by ArcGIS 10.2 based on a 250 m × 250 m grid unit. Second, the ecological footprint and ecological carrying capacity of cultivated land were assessed by the value of ecosystem services calculated by the functional value method. Third, the calculating method of farmland yield factor in ecological carrying capacity for cultivated land was developed, which was introduced in this study to improve the accuracy of the results. Then, we developed an ecological equilibrium index to identify whether the ecological payment area or the compensation zone based on comparing the ecological footprint with ecological carrying capacity was a better option. Finally, the value of horizontal ecological compensation for cultivated land was calculated on account of the development conditions of society and the economy. Jiangxi province, as one of 13 provinces of major grain production areas in China, is where the cultivated land ecosystem was relatively stable and utilization was rather high. However, the quantity, quality and ecological protection of cultivated land has become increasingly prominent in Jiangxi along with economic development, urbanization, acceleration of industrialization, and the rapid expansion of urban population in recent years. Discussing the horizontal ecological compensation for cultivated land within the region has important theoretical and practical significance for the improvement of cultivated land quality and the balance between economic development and ecosystems on a regional scale.

## 2. Materials and Methods

### 2.1. Study Area

Jiangxi province is situated in the middle reaches of the Yangtze River in China. It is also one of the areas of food allotment, ranked second in the provinces to the south of Yangtze River. In 2021, the total area of cultivated land was 2,721,600 hm^2^ in Jiangxi province, and paddy field accounted for approximately 2,270,358 hm^2^ (83.42%) of the total cultivated land in this area, which was mainly distributed in the hills, intermontane or valley basins. The gross value of agricultural output of Jiangxi in 2021 was CNY 3998.1 billion. Its total grain output was 219.25 million tons, and above 2.15 million tons in the 9 successive years after 2012.

### 2.2. Data Sources

The data in this study were mainly from satellite remote-sensing by GIS technology processing. The detailed sources are as follows: the data of natural geography came from the National Earth System Science Data Center (http://www.geodata.cn, accessed on 10 January 2022), including cultivated land use area and net primary production of farmland plant(NPP); soil erodibility factor and factor of length and degree of slope and the soil quality data, including nitrogen, phosphorus, potassium and organic matter in soil sample, were obtained from sample data results of soil testing and fertilizer recommendation in Jiangxi agriculture; the meteorological data, including average temperature, annual precipitation and annual potential evaporation in 2018, were from Resource and Environment Science and Data Center (http://www.resdc.cn, accessed on 10 January 2022) and the MOD16A2-PET product of NASA (https://ladsweb.modaps.eosdis.nasa.gov, accessed on 10 January 2022); the prices of various food crops and the unit prices of pesticides and fertilizers were derived from the Compilation of National Agricultural Product Costs and Benefits; the socio-economic development data, including the data of population and grain production and consumption, were taken from the Jiangxi Statistical Yearbook 2019, Economic and the Social Development Statistical Bulletin 2019 and historical Statistical Yearbooks of 11 cities of Jiangxi in 2018.

### 2.3. Methods

#### 2.3.1. Measurement of Ecosystem Service Function Value

The ecosystem service value was composed of the providing, regulating, cultural, and supporting service values (Costanza et al., 1997) [35]; however, the providing and cultural values have been accounted for in the Chinese SNA. In order to avoid repeated accounting, the main focus in this study is on the regulating and supporting service value. According to the previous studies, it can be inferred that the ecosystem service value including soil conservation, water conservation, carbon fixation and oxygen release were regarded as the most important ecological service values in agroecosystem [36,37,38], and the calculation method of these three values are of very wide application and validity. Thus, the amount of ecosystem service functions of soil conservation, water conservation, carbon fixation and oxygen release were calculated by the functional value method. Then, its value-standardized calculation can be achieved by the surrogate market, shadow project, carbon tax, and the production cost method. The calculation formula can be expressed as follows:(1)$$E_{v}=V_{es}+V_{ew}+V_{ea}$$
where $E_{v}$ is the total value of ecosystem service function for cultivated land; $V_{es}$ is the value of soil conservation service function; $V_{ew}$ is the value of water conservation service function; $V_{ea}$ is the value of carbon fixation and oxygen release service function.

(1)The value of soil conservation service function

The soil conservation service is the function that slows down the loss of soil fertility due to soil erosion. The amount of soil conservation was calculated by the Universal Soil Loss Equation (USLE), and the value of soil conservation service can be achieved by the surrogate market. The calculation formula can be expressed as
(2)$$Q_{ser}=R\times K\times LS\times C\times P$$
(3)$$Q_{sep}=R\times K\times LS$$
(4)$$Q_{sm}=Q_{sep}-Q_{ser}$$
(5)$$V_{es}=\sum Q_{sm}\times Q_{ei}\times P_{ei}\left(i=N,\;P,\;K,\;M\right)$$
where $Q_{ser}$ is the actual amount of soil erosion for cultivated land; R is the data of rainfall erosivity; K is the data of soil erodibility factor; LS is the data of length and degree of slope; C is the data of vegetation cover factor, its value is 0.23 which comes from a previous study (Wang et al., 2019) [39]; P is the data of soil conservation factor, its value of paddy field is 0.01 and that dry land is 0.4, which comes from a previous study (Liu et al., 2018) [40]; $Q_{sep}$ is the data of potential soil erosion; $Q_{sm}$ is the data of soil conservation quantity by means of USLE(t) [41]; $V_{es}$ is the value of soil conservation service; $Q_{ei}$ is the contents of nitrogen, phosphorus and potassium in soil samples; $P_{ei}$ is the market price of chemical fertilizer.

(2)The value of water conservation function

The water conservation service is the function that regulates surface runoff and supplements the groundwater lost by water evaporation for the farmland ecosystem. The amount of water conservation was calculated by the Water Balance Method (Zeng et al., 2014) [42], and the value of water conservation service can be achieved by shadow engineering approach. The calculation formula can be expressed as follows:(6)WY=P−ET
(7)$$ET=\frac{P\left(1+\omega \frac{PET}{P}\right)}{1+\omega \frac{PET}{P}+{\left(\frac{PET}{P}\right)}^{-1}}$$
(8)$$V_{ew}=WY\times P_{w}\times A_{i}$$
where WY is the amount of water conservation function for cultivated land; P is the amount of annual precipitation (mm); ET is the total amount of discharge (mm); ω is the land cover factor; PET is the amount of annual evapotranspiration (mm); $V_{ew}$ is the value of the water conservation function; $P_{w}$ is the manufacturing cost of reservoir, its value is CNY 0.67 per cubic meter, which comes from the author’s previous study (Zhong, 2022) [33]; $A_{i}$ is the amount of cultivated land.

(3)The value of carbon fixation and oxygen release function

The carbon fixation and oxygen release service is the function that stores carbon and produces oxygen by soil, vegetation and germs for farmland ecosystem. The crops could fix 1.63 g CO^2^ and release 1.19 g O^2^ with the formation of 1 g of dry matter according to the photosynthesis equation (Wang et al., 2018) [43]. The calculation formula can be expressed as follows:(9)$$V_{ea}=U_{c}+E_{o}$$
(10)$$U_{c}=1.63\times R_{c}\times NPP\times f_{c}$$
(11)$$E_{o}=1.19\times NPP\times C_{o}$$
where $V_{ea}$ is the total amount of the carbon fixation and oxygen release function for cultivated land (CNY/m^2^); $U_{c}$ is the amount of carbon fixation for cultivated land; $E_{o}$ is the amount of the oxygen release function for cultivated land; $R_{c}$ is the carbon content of CO^2^, its value is 27.27% which comes from a previous study(Xue et al., 2020) [44]; $f_{c}$ is the price of carbon fixation for cultivated land (Xue et al., 2020) [44]; NPP represents the data of net primary production of farmland plants (g C/m^2^); $C_{o}$ is the cost of oxygen production (Xue et al., 2020) [44].

#### 2.3.2. Measurement of Ecological Footprint

The ecological footprint of farmland is the amount of actual occupation of cultivated land based on food demand for population in certain areas. The calculation formula can be expressed as follows:(12)$$E_{f}=N\times ef$$
(13)$$ef=\alpha \times \overset{n}{\underset{i=1}{\sum }}r\frac{c_{i}}{p_{i}}$$
where $E_{f}$ is the total amount of ecological footprint for cultivated land in certain areas (hm^2^); N is the total amount of permanent residents in certain areas (per person); ef is the amount of per capita ecological footprint for cultivated land (hm^2^/person); n is the category of consumer goods for cultivated land in certain areas, its value is 3, that is the staple crops including rice, soybean and sweet potato; i is the specific agricultural products supplied by cultivated land; r is the equivalence factor of cultivated land in certain areas; Ci is the per capita consumption of i type in certain areas (kg); Pi is the annual average output in Jiangxi of i type crops; α is the per capita consumption of grain crops and all crops ratio in Jiangxi, its value is 2.26 (Ruan et al., 2021) [45].

#### 2.3.3. Measurement of Ecological Carrying Capacity

The ecological carrying capacity is the ability for land to provide natural supplies for humans to survive and develop. The ecological carrying capacity of farmland is mainly reflected in the supply capacity of grain crops to meet the demand of human survival by cultivated land as an essential productive factor. The calculation formula can be expressed as
(14)$$E_{c}=N\times a\times r\times y\times \left(1-12\%\right)$$
where $E_{c}$ is the data of ecological carrying capacity for cultivated land (hm^2^); N is the total amount of permanent residents in certain areas (per person); a is the per capita occupied amount of cultivated land; r is the equivalence factor of cultivated land in certain areas; y is the yield factor of cultivated land in certain areas. In addition, 12% of the biologically productive area should be deducted for the protection of biodiversity in the area according to the United Nations World Commission on Environment and Development (WCED).

Moreover, the farmland yield factor is based on the previous studies [46,47], which were widely quoted. Their results of the farmland yield factor are applied only to ecological carrying capacity at the macro-scale, such as national and provincial scales, but are not the appropriate method for calculation of ecological carrying capacity at small and medium scales. In order to improve the accuracy of results, this study attempted to calculate the data of farmland yield factor at city scale in Jiangxi using the research method of Liu et al. (2010) [47]. The results of farmland yield factors are listed in Table 1.

#### 2.3.4. Measurement of Ecological Profit/Loss and Equilibrium Index

The ecological profit/loss is the difference between ecological carrying capacity and ecological footprint of farmland. The equilibrium index is mainly reflected the sustainability status of ecological environment for cultivated land by ratio of ecological carrying capacity to sum of ecological carrying capacity and ecological footprint of farmland.
(15)$$E_{p}=E_{c}-E_{f}$$
(16)$$E_{b}=\frac{E_{c}}{E_{c}+E_{f}}$$
where $E_{p}$ is the data of ecological surplus and deficit for cultivated land (hm^2^); $E_{b}$ is the index of ecological balance for cultivated land, when $E_{b}$ > 0.5, it indicates the region is in the state of ecological surplus. By contrast, when $E_{b}$ < 0.5, it indicates the region is in the state of ecological deficit.

#### 2.3.5. Calculation of Horizontal Ecological Compensation Value

The calculation of a reasonable ecological compensation value needs to consider not only ecological benefit of cultivated land, but also the utilization situation of cultivated land and the local socio-economic situation. The amount of horizontal ecological compensation value was calculated by the amount of ecosystem service function value, ecological equilibrium index and coefficient of correction, which counted by the social and economic development conditions in Jiangxi according to the previous research (Chen et al., 2007) [48]. The calculation formula can be expressed as follows:(17)$$E_{s}=E_{b}\times E_{v}\times k$$
(18)$$k=\frac{1}{1+e^{-\left(e_{1}+e_{2}\right)/2}}$$
where $E_{s}$ is the total amount of ecological compensation value for cultivated land (CNY); $E_{b}$ is the index of ecological balance for cultivated land; $E_{v}$ is the total value of ecosystem service function for cultivated land; k is the correction factor; $e_{1}$ and $e_{2}$ are Engel’s coefficients of urban and rural in China, respectively.

## 3. Results and Analysis

### 3.1. Spatial Changes of Ecosystem Service Function Value for Cultivated Land

The total value of ecosystem service function for cultivated land in Jiangxi was CNY 148.68 billion in 2018. The value of soil conservation service, CNY 88.69 billion, exceeds others, accounting for 59.65% of the total value of ecosystem services of cultivated land in Jiangxi, as shown in Table 2. From a unit value perspective, the unit value of the ecosystem service function presented a distribution pattern of “increasing from the center on Poyang Lake Basin to the periphery gradually” in Jiangxi province, and was coincident with the distribution of soil conservation, carbon fixation and oxygen release service value per unit. In contrast, the water conservation service value per unit presented a decreasing shape from the center to the surroundings in Jiangxi, as shown in Figure 1. Ganzhou City has the most value per unit of arable land ecosystem service from other cities with 47,243.31 CNY/hm^2^, followed by Pingxiang (44,991.51 CNY/hm^2^), Jiujiang (41,406.39 CNY/hm^2^), Jingdezhen (39,555.85 CNY/hm^2^), Shangrao (38,465.76 CNY/hm^2^), Fuzhou (32,005.38 CNY/hm^2^), Ji’an (31,567.05 CNY/hm^2^), Yichun (29,235.37 CNY/hm^2^), Yingtan (27,514.48 CNY/hm^2^), Xinyu (26,085.81 CNY/hm^2^) and Nanchang (15,281.73 CNY/hm^2^). From the total value perspective, Ganzhou City is still the highest value of arable land ecosystem service from other cities with CNY 30.18 billion, followed by Shangrao (CNY 23.11 billion), Jiujiang (CNY 21.41 billion), Ji’an (CNY 20.38 billion), Yichun (CNY 17.99 billion), Fuzhou (CNY 14.98 billion), Nanchang (CNY 5.80 billion), Jingdezhen (CNY 4.73 billion), Pingxiang (CNY 3.97 billion), Yingtan (CNY 3.13 billion) and Xinyu (CNY 2.95 billion), as shown in Table 2 and Figure 2.

### 3.2. The Results of Ecological Footprint and Ecological Carrying Capacity for Cultivated Land

Calculating the ecological profit and loss of cultivated land in cities of Jiangxi province can effectively identify the ecological surplus and deficit of cultivated land according to Formula (15). Nanchang, Pingxiang and Jiujiang were the cultivated land ecological deficit areas in 2018, since the population density and industrial economic development of the three cities was in the forefront in all cities in Jiangxi, located in Poyang Lake Economic Belt. However, these cities were also subject to payment areas of horizontal ecological compensation for cultivated land in Jiangxi due to weak grain production capacity. By contrast, eight cities including Jingdezhen, Xinyu, Yingtan, Ganzhou, Ji’an, Yichun, Fuzhou and Shangrao, are arable land ecological surplus regions, which have larger arable land areas and favorable conditions for agricultural cultivation. Moreover, the three cities with the highest ecological surplus areas were Yichun, Jian and Fuzhou, and the ecological surplus areas of these cities is 853.3 thousand hm^2^, up to about 67.78% ecological surplus area of all cities in Jiangxi, as shown in Table 3 and Figure 3.

### 3.3. The Results of Horizontal Ecological Compensation Values for Cultivated Land in Jiangxi

The values of horizontal ecological compensation for cultivated land in the 11 cities of Jiangxi province were calculated according to Formula (17). From the amount of ecological compensation perspective, the three cities with the highest amount of ecological compensation were Ganzhou, Shangrao and Ji’an, and the total amount of compensation in these cities is CNY 25.14 billion, making up about 50.83% of the horizontal ecological compensation values of all cities in Jiangxi, as shown in Figure 4. From the values of ecological profit and loss perspective, payment districts are either the districts with relatively small amounts of farmland and ecological carrying capacity, such as Pingxiang (CNY −1.09 billion) and Nanchang (CNY −1.52 billion), or the districts in which the ecological footprint is relatively higher, e.g., Jiujiang (CNY 5.33 billion). Meanwhile, other cities were the ecological compensation zones, and the total values of ecological compensation in these cities (CNY 41.52 billion) is nearly 5.5 times of the total payment values (CNY 7.45 billion).

The reasons for the results are mainly attributed to the calculation method of horizontal ecological compensation based on an improved ecological footprint model, and the ecological supply capacity of cultivated land in Jiangxi was much above the national average. However, because it is difficult to achieve the absolute equilibrium of horizontal ecological compensation for cultivated land within Jiangxi province, the difference between compensation and payment should be made up for by the central and provincial budgets.

## 4. Discussion

### 4.1. The Rationality of the Results

This study attempted to assess the value of horizontal ecological compensation of cultivated land based on an improved ecological footprint model, and unlike some traditional methods of ecosystem service value [49,50,51], it focused on not only the capacity of ecosystem service in farmland, but also the socio-economic situation. In order to verify the rationality of the value of horizontal ecological compensation, the cost of cultivated land protection and public expenditure burden should be compared with the results of ecological compensation value in our study. First, as shown in Table 4 and Table 5, the unit value of horizontal ecological compensation for cultivated land in 11 cities of Jiangxi is nearly 2.3~8.7 times of the cost of cultivated land protection, which was described as the government subsidies to inspire agrarian willingness to protect cultivated land by Cai’s study [52]. This indicated that the results will play the driving role in the protective behavior for cultivated land because the value of horizontal ecological compensation is higher than the cost of cultivated land protection. Second, a reasonable result of ecological compensation value should not be a heavy financial burden on the government. From the ecological deficit areas perspective, the share of ecological compensation values of the total fiscal revenue in the three cities (Nanchang, Jiujiang and Pingxiang) has stay at a low level; at less than 11%, it can give sufficient financial support to the ecological surplus areas because of its higher GDP and make the larger payment amounts with less financial pressure. Conversely, the proportion of ecological compensation values in the ecological surplus areas is higher than ecological deficit areas, but is still less than 29%. The results shows that the value of horizontal ecological compensation can still be effective to encourage protective behavior for arable land, as shown in Table 5. Moreover, the share of ecological compensation values in the agricultural fiscal expenditure presents more differences between the ecological surplus areas and deficit areas. The maximum proportion was up to 119.59%, which indicated that ecological compensation values are an important source of agricultural fiscal expenditure and help to promote local agricultural development.

In addition, the unit value of horizontal ecological compensation of cultivated land in Jiangxi (7806.78 CNY/hm^2^) is more in line with the actual situation in Chinese ecological compensation standards, compared with the results of Chengdu (4500~6000 CNY/hm^2^), Suzhou (6000~9000 CNY/hm^2^) and Hangzhou (6000~12,000 CNY/hm^2^) [53].

### 4.2. Comparison with the Literature

The unit value of horizontal ecological compensation of cultivated land in Jiangxi province in our study is estimated at 7806.78 CNY/hm^2^, which was much smaller than the estimate of 19,439.98 CNY/hm^2^ in the study by Liu et al. (2022) [34]. A different method to quantify the value of cultivated land ecosystem services is the main reason for the difference; namely, Liu et al. (2022) [34] calculated the value of cultivated land ecosystem services by the equivalent factor method, while the functional value method was used in our study. However, the unit value in our study was 7806.78 CNY/hm^2^ in 2018, almost 1.1 times the estimate of 7104.11 CNY/hm^2^ in the study by Zhang et al. (2020) [54] in 2017. This difference can be fully explained by the different research periods and calculation methods of ecosystem service values. Compared with the other existing studies [4,45,55], the ecological compensation standard of cultivated land per unit calculated in this study is relatively lower, mainly due to three aspects. First, the ecological compensation standard in our study is used for the balance between economic development and ecosystem on the city scale in Jiangxi province, yet the provincial balance was the research goal in most other studies. Second, our study considers the different ecosystem service functions of cultivated land, and fully displays the values of both ecosystem service functions on grid scale using the functional value method. Third, the other existing studies estimated the value of ecological compensation based on the market value of cultivated land, but the cost and expense of maintaining ecosystem services in cultivated land was used for the calculation of ecological services value in our study; therefore, the value of the ecological compensation in cultivated land per unit is relatively lower.

Last but not least, both the biological capacity and ecological footprint in our study used in the ecosystem service calculation rely on large amounts of basic data such as the soil content data including nitrogen, phosphorus and potassium from 2018. However, the latest data of soil content was not available. Therefore, this study has only discussed the spatial change in Jiangxi in 2018, and has not analyzed the temporal change in values of ecosystem services and the ecological footprint. In the future, we will further analyze the relation including soil content and ecological footprint when the soil data can be updated in time and improve the accuracy of horizontal ecological compensation.

## 5. Conclusions

In this study, the ecological compensation standard of cultivated land in each city in Jiangxi were calculated by an improved ecological footprint model, which was based on the functional value method of ecosystem services using a 250 m × 250 m grid unit. It was reliable to use the results of ecological compensation standard to delimit the ecological deficit and surplus areas of cultivated land. The main conclusions are as follows:(1)The cities that need to pay the ecological compensation for cultivated land are Nanchang, Jiujiang and Pingxiang because their ecological footprints are greater than their carrying capacities, at the total amount of CNY 7.94 billion. By contrast, other cities in Jiangxi were the ecological surplus zones, at the total amount of CNY 41.52 billion. This shows that the buyers and suppliers of ecosystem services for cultivated land in Jiangxi can be distinguished using the method in our study.(2)The ecological payment areas of cultivated land were generally distributed in the northwest of Jiangxi and the surrounding Poyang Lake, because these areas have higher population densities and faster development of industrial economies. It shows that the ecological situation of arable land is not only related with the natural condition in Jiangxi, but also with the socio-economic status closely.(3)The number of cities and the total ecological compensation values in ecological surplus areas were greater than the total of ecological deficit areas. This also indicated that there is a large amount of arable land, which is a favorable condition for agricultural cultivation. It has a better supply capacity for ecosystem services in most of the cites in Jiangxi, which is one of the thirteen major grain-cultivating areas in China.(4)The results of this study could inspire the local government to continue to protect cultivated land because the values of ecological compensation are higher than the cost of farmland protection. Moreover, the payment amount of ecological compensation is affordable for the government to finance and contribute to the balance between regional economy and ecology.

## Figures and Tables

**Figure 1 ijerph-20-04618-f001:**
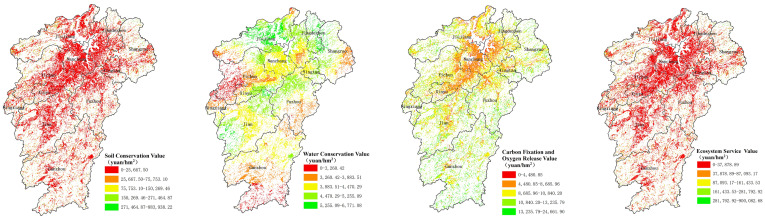
Spatial distribution map of unit price of cultivated land resource asset ecosystem service function value.

**Figure 2 ijerph-20-04618-f002:**
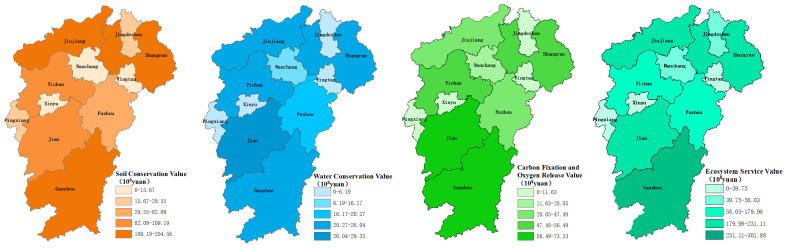
Spatial distribution map of total value of cultivated land resource asset ecosystem service function.

**Figure 3 ijerph-20-04618-f003:**
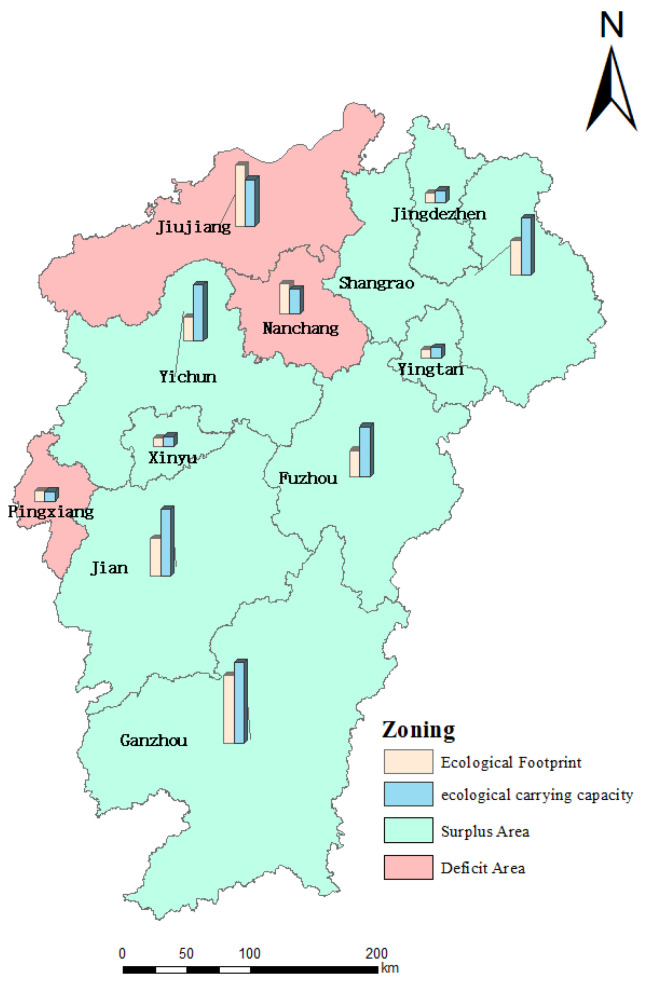
Spatial distribution of ecological surplus of cultivated land resource assets.

**Figure 4 ijerph-20-04618-f004:**
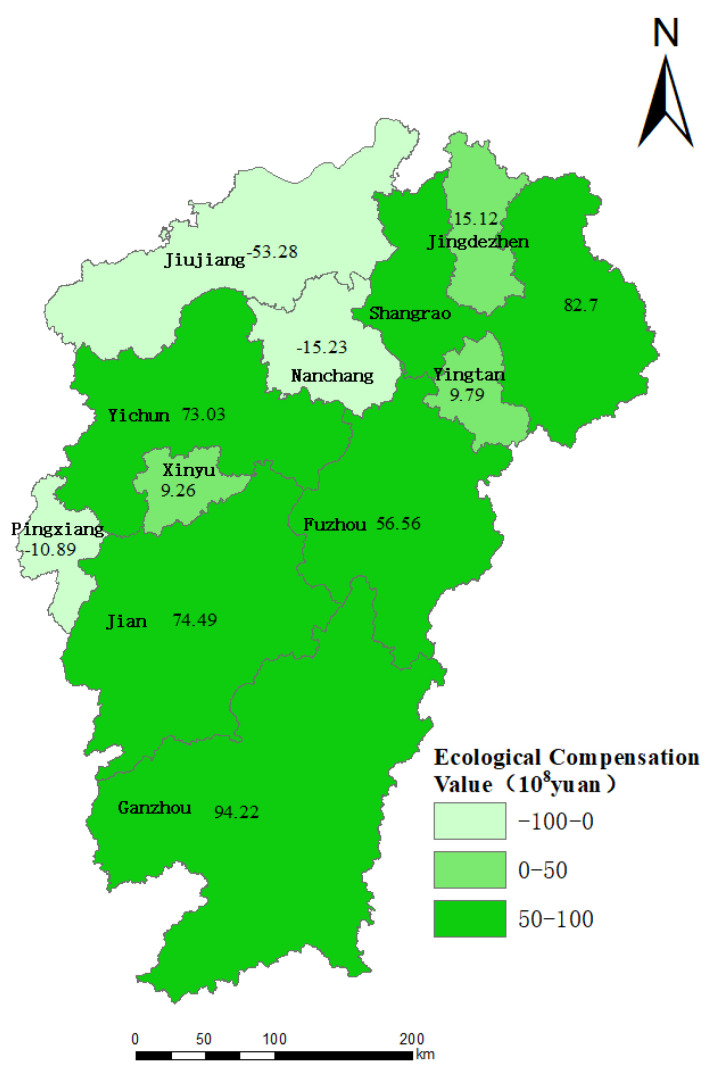
Spatial distribution map of total amount of ecological compensation standards for cultivated land resource assets.

**Table 1 ijerph-20-04618-t001:** Farmland yield factors in various cities of Jiangxi province.

City	NPP (g C/m^2^/Year)	Farmland Yield Factors
Ganzhou	1489.06	1.28
Pinxiang	1279.05	1.10
Jiujiang	1059.73	0.91
Jingdezhen	1221.66	1.05
Shangrao	1120.09	0.96
Fuzhou	1253.07	1.08
Jian	1218.78	1.05
Yichun	1069.53	0.92
Yingtan	1103.57	0.95
Xinyu	1047.41	0.90
Nanchang	782.13	0.67
Summation	1164.16	—

**Table 2 ijerph-20-04618-t002:** The value of cultivated land resource assets by ecosystem service function.

City	Area (10^3^ hm^2^)	SC-Value (CNY Billion)	WC-Value (CNY Billion)	CFOR-Value (CNY Billion)	Total Value (CNY Billion)	(%)
Nanchang	379.80	1.38	1.62	2.81	5.80	3.90%
Jingdezhen	119.78	2.96	0.62	1.16	4.74	3.19%
Pingxiang	88.35	2.75	0.33	0.90	3.98	2.67%
Jiujiang	517.07	14.09	2.57	4.75	21.41	14.40%
Xinyu	113.28	1.47	0.47	1.01	2.96	1.99%
Yingtan	113.94	1.57	0.50	1.07	3.14	2.11%
Ganzhou	638.99	20.46	2.38	7.35	30.19	20.30%
Jian	645.80	10.92	2.94	6.53	20.39	13.71%
Yichun	615.59	10.01	2.34	5.65	18.00	12.10%
Fuzhou	468.14	8.21	2.03	4.75	14.98	10.08%
Shangrao	600.82	14.89	2.61	5.62	23.11	15.54%
Total	4301.56	88.69	18.40	41.60	148.68	—

**Table 3 ijerph-20-04618-t003:** Ecological surplus of cultivated land resource assets in various cities of Jiangxi province.

City	Area (10^3^ hm^2^)	Inhabitant (10^3^ Person)	Ecological Footprint (10^3^ hm^2^)	Ecological Carrying Capacity (10^3^ hm^2^)	Profit and Loss Area (10^3^ hm^2^)	Compensation Zone
Nanchang	379.80	5545.54	298.36	251.49	−46.86	deficit
Jingdezhen	119.78	1673.21	98.80	123.88	25.07	surplus
Pingxiang	88.35	1933.15	104.83	95.73	−9.10	deficit
Jiujiang	517.07	4896.85	605.50	463.70	−141.81	deficit
Xinyu	113.28	1186.73	83.56	100.41	16.85	surplus
Yingtan	113.94	1175.01	89.35	106.65	17.30	surplus
Ganzhou	638.99	8677.60	675.95	806.02	130.06	surplus
Ji’an	645.80	4956.57	381.24	668.33	287.08	surplus
Yichun	615.59	5573.24	231.11	558.11	327.00	surplus
Fuzhou	468.14	4047.15	259.09	498.27	239.18	surplus
Shangrao	600.82	6810.66	343.16	568.46	225.30	surplus
Total	4301.56	46,475.73	3190.18	4239.07	1048.89	—

**Table 4 ijerph-20-04618-t004:** The socio-economic situation in Jiangxi province.

City	GDP (CNY Billion)	Fiscal Revenue (CNY Billion)	Agricultural Fiscal Expenditure (CNY Billion)
Nanchang	527.47	86.94	4.30
Jingdezhen	84.66	13.25	1.39
Pingxiang	100.91	16.16	1.96
Jiujiang	270.02	50.81	5.53
Xinyu	102.73	14.43	1.38
Yingtan	81.90	14.09	1.13
Ganzhou	280.72	45.95	9.15
Ji’an	174.22	28.05	6.23
Yichun	218.09	39.14	6.39
Fuzhou	138.24	20.07	5.67
Shangrao	221.28	35.16	7.27
Total	2200.23	364.02	50.39

**Table 5 ijerph-20-04618-t005:** The rationality analysis of ecological compensation values in Jiangxi province.

City	The Total Values of Ecological Compensation (CNY Billion)	The Unit Value of Horizontal Ecological Compensation (CNY/hm^2^)	The Proportion of Fiscal Revenue (%)	The Proportion of Agricultural Fiscal Expenditure (%)
Nanchang	−1.52	−4010.01	−1.75	−35.43
Jingdezhen	1.51	12,631.58	11.41	108.91
Pingxiang	−1.09	−12,332.96	−6.74	−55.44
Jiujiang	−5.33	−10,305.61	−10.49	−96.36
Xinyu	0.93	8180.21	6.42	67.33
Yingtan	0.98	8595.26	6.95	86.83
Ganzhou	9.42	14,747.22	20.50	102.97
Ji’an	7.45	11,534.53	26.55	119.59
Yichun	7.30	11,865.15	18.66	114.27
Fuzhou	5.66	12,082.89	28.18	99.83
Shangrao	8.27	13,764.98	23.52	113.68
Total	33.58	7806.78	9.22	66.63

## Data Availability

Not applicable.

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
