# Peer review of "Quantitative Assessment of Horizontal Ecological Compensation for Cultivated Land Based on an Improved Ecological Footprint Model: A Case Study of Jiangxi Province, China"

_ijerph, 2023, doi:10.3390/ijerph20054618_

Round 1

Reviewer 1 Report

I have completed the review of manuscript submitted by Zhong et al. Authors have undertaken an extensive assessment of horizontal ecological compensation in several cities of Jiangxi province, China, with an aim to have a better coordinated model based on improved ecological foot print methods. The study is meticulously planned and PES has been investigated using several parameters and indices. The work is worthy of publication. However, there are few points that need attention:

1. What was the hypothesis /rationale for undertaking the project?

2. How the projections based on the study help in determining PES in other parts of the world? Is there any role of variability in edaphic/topographic/climatic/ economic conditions?

3. Why did the authors choose this province?

4. How could the findings of the study help in modulating/altering peoples' perception about PES?

5. The abstract should be revised to provide all information about the findings of the study. The cities selected in the study should be explicitly explained in the abstract. How are these findings better than the previous findings?

6. All these aspects need to be explained/discussed in the manuscript. 

Author Response

Point 1: What was the hypothesis /rationale for undertaking the project?

Response 1: Thank you for your valuable comments. The ecological footprint model are more helpful to reflect ecological benefits for farmland in current researches, and many studies estimated ecosystem service value for cultivated land in the ecological footprint model using the equivalent factor method, which constructed a standard and objective equivalent factor table,  but the fixed coefficient of equivalent factor could hardly reflect the spatial heterogeneity of different ecosystem service functions accurately due to ecosystem service function is dynamic. So this study attempted to assess the ecosystem service value for cultivated land in the ecological footprint model using the functional value method, which could reflected the dynamic change of spatio-temporal heterogeneity in ecosystem service at a regional scale more accurate.(Line 79-107, Page 2-3)

Point 2: How the projections based on the study help in determining PES in other parts of the world? Is there any role of variability in edaphic/topographic/climatic/ economic conditions?

Response 2: Thank you for your valuable comments. We have revised the expression of data sources in our study(Line 123-137, Page 3). The advantage of the functional value method in the study is able to reflect the dynamic change of spatio-temporal heterogeneity in ecosystem service more accurate, and the data of edaphic quality, climatic conditions and topography in the study area were mainly from satellite remote-sensing and GIS technology. The data of economic conditions in the study were obtained from the Statistical Yearbook of China. Therefore, the PES in other parts of the world should be precisely calculated by the method of our study in theory.

Point 3: Why did the authors choose this province?

Response 3: Thank you for your valuable comments. We agree that this point is important, and we have explained in detail the reason of Jiangxi as the research area. (Line 103-108, Page 3)

Point 4: How could the findings of the study help in modulating/altering peoples' perception about PES?

Response 4: Thank you for your valuable comments. It is widely accepted that PES has been applied to basin, water rights trading, pollutant emissions trading, and carbon emissions trading. But it has not yet been implemented in the horizontal ecological compensation for cultivated land in major grain cultivating areas in China. So we have revised the expression of contributions of the study’s finding in conclusion.(Line 391-408, Page 13)

Point 5: The abstract should be revised to provide all information about the findings of the study. The cities selected in the study should be explicitly explained in the abstract. How are these findings better than the previous findings?

Response 5: Thank you for your valuable comments. We agree that this point is important, and we have explained in detail the reason of Jiangxi selected and all information about the findings of the study in our abstract.(Line 18-33, Page 1)

Point 6: All these aspects need to be explained/discussed in the manuscript.

Response 6: Thank you for your valuable comments. We have carefully considered all issues raised by you and responded specifically to each of your comments (in red). We hope that the corrections made will meet your expectations.

Reviewer 2 Report

A lot of research has been done on the horizontal ecological compensation of cultivated land. Although your work has carefully considered the value of various ecosystem services of cultivated land, the model in this paper lacks theoretical support.

The theory of ecosystem service payment studies the issue of ecological compensation from the perspective of beneficiary compensation, without considering the balance of ecosystem system, unless it can be constructed that the decrease of beneficiary payment is due to the increase of ecological footprint. The author can refer to the research of Ding and Yao(2019,2022)`s researches. The core of ecological compensation is to calculate the price of ecosystem services, not the total value amount.

 The model without price is meaningless. The model (17) is hard to calculate the value of the ecological deficit area. Should the formula be changed to 

Literature 33 is an unpublished document and quoted by yourself. And the cost of Uc and E0 is doubtful.

I hope this paper can find a more reasonable theory to solve the above problems.

Author Response

Point 1: A lot of research has been done on the horizontal ecological compensation of cultivated land. Although your work has carefully considered the value of various ecosystem services of cultivated land, the model in this paper lacks theoretical support.

Response 1: Thank you very much for your suggestion. The principal theory of our study is ecological footprint theory, and the ecological footprint and ecological carrying capacity of cultivated land were assessed by the value of ecosystem services. In our study, we developed ecological equilibrium index to identify whether ecological payment area or compensation zone based on compared ecological footprint with ecological carrying capacity, which consider the balance of ecosystem system. Moreover, our study attempted to assess the ecosystem service value for cultivated land in the ecological footprint model using the functional value method, which could reflected the dynamic change of spatio-temporal heterogeneity in ecosystem service at a regional scale more accurate.(Lines88-107,Page 2-3)

Point 2: The theory of ecosystem service payment studies the issue of ecological compensation from the perspective of beneficiary compensation, without considering the balance of ecosystem system, unless it can be constructed that the decrease of beneficiary payment is due to the increase of ecological footprint. The author can refer to the research of Ding and Yao(2019,2022)`s researches. The core of ecological compensation is to calculate the price of ecosystem services, not the total value amount.

Response 2: Thank you very much for your suggestion. We agree that this point is important. We explained the concept of ecological footprint theory in detail, and cited new references to supplement the research theory(Lines42-57,Page 2). Moreover, the model of ecological footprint with ecological carrying capacity in our study is similar with the research of Ding and Yao(2019,2022) basically, and the main difference is the method of calculation, namely, Ding and Yao(2019,2022) calculated the value of cultivated land ecosystem services by equivalent factor method, while the functional value method was used in our study(Lines79-91,Page 2). But I will be devoted to study the model by considering vertical and horizontal compensation for cultivated land in future study according to your suggestions.

Point 3: The model without price is meaningless. The model (17) is hard to calculate the value of the ecological deficit area. Should the formula be changed to.

Response 3: Thank you for your valuable comments. We have carefully checked this section, it may be that the expression has caused a misunderstanding. So we have revised the unit of ES(Lines 248, Page 7) and cited reference to explain the contents of model in detail(17). (Lines 239-244, Page 7)

Point 4: Literature 33 is an unpublished document and quoted by yourself. And the cost of Uc and E0 is doubtful..

Response 4: Thank you for your valuable comments. So, we have replaced "my Ph.D dissertation" with "the research of (Xue et al.,2020)" in the text.(Lines 191-193, Page 5)

Point 5: I hope this paper can find a more reasonable theory to solve the above problems..

Response 5: Thank you for your valuable comments. We have carefully considered all issues raised by you and responded specifically to each of your comments (in red). We hope that the corrections made will meet your expectations.

Round 2

Reviewer 2 Report

No others qustions